# Patient-Specific Circulating Tumor DNA for Monitoring Response to Menin Inhibitor Treatment in Preclinical Models of Infant Leukemia

**DOI:** 10.3390/cancers16233990

**Published:** 2024-11-28

**Authors:** Louise Doculara, Kathryn Evans, J. Justin Gooding, Narges Bayat, Richard B. Lock

**Affiliations:** 1Children’s Cancer Institute, Lowy Cancer Research Centre, School of Clinical Medicine, UNSW Medicine & Health, UNSW Centre for Childhood Cancer Research, UNSW Sydney, Sydney, NSW 2052, Australia; ldoculara@ccia.org.au (L.D.); nbayat@ccia.org.au (N.B.); 2School of Chemistry, UNSW Sydney, Sydney, NSW 2052, Australia; 3Australian Centre for NanoMedicine, UNSW Sydney, Sydney, NSW 2052, Australia

**Keywords:** circulating tumor DNA, liquid biopsy, patient-derived xenograft, minimal residual disease, menin inhibitor

## Abstract

Children with aggressive acute lymphoblastic leukemia (ALL) currently undergo frequent, invasive bone marrow biopsies to monitor their treatment response. This research explores a less invasive alternative using circulating tumor DNA (ctDNA) in blood plasma. We used a mouse model of ALL to study how ctDNA levels change during leukemia progression and treatment. Our findings show that ctDNA can accurately reflect the spread of leukemia and may be better at detecting the presence of residual cancer cells in organs compared to traditional blood tests. This research could lead to less invasive and more effective ways of monitoring treatment response in young leukemia patients, potentially improving their care and outcomes.

## 1. Introduction

Acute lymphoblastic leukemia (ALL) is the most common childhood cancer [1]. Despite improvements in cure rates, the survival remains poor for patients who relapse or have chemotherapy-resistant disease [2]. One aggressive ALL subtype is *KMT2A* (*MLL1*)-rearranged (MLL-r) ALL, which occurs in ~80% of infants with ALL [3]. The strongest prognostic factor for increased risk of relapse is the persistence of leukemia cells during and post therapy at levels below morphologic detection in the bone marrow (BM, minimal/measurable residual disease, MRD). Therefore, early detection of MRD is critical for evaluating initial treatment response and adjusting treatment protocols [4,5,6].

Currently, MRD is monitored using repeated BM biopsies followed by either real-time quantitative PCR (qPCR) detection of specific clonal rearrangements of immunoglobulin (Ig) and T cell receptor (TCR) genes or detection of MRD cells by flow cytometry [4,7]. However, the sensitivity of current MRD testing methods is limited by the amount of sample that can be collected from invasive BM biopsy at any time point [5,6]. Also, relapses occurring in extramedullary sites cannot be predicted by BM biopsy [8]. Therefore, techniques that can detect MRD in the blood with higher sensitivity are urgently needed to replace BM biopsy and to improve MRD testing and patient outcome.

In recent years, the analysis of cell-free DNA (cfDNA) and circulating tumor DNA (ctDNA) in blood plasma has gained attraction as a non-invasive method for early disease detection, allowing for more frequent sampling and molecular monitoring. CfDNA consists of DNA fragments in the bloodstream originating from various cellular processes including apoptosis, and cfDNA levels often increase in disease states. The utility of total cfDNA quantification in disease monitoring is debatable, primarily due to uncertainties surrounding its origin, clearance mechanisms, and non-standardized measurement protocols [9]. In contrast, ctDNA, a subset of cfDNA, specifically originating from tumor cells, offers a more accurate measure of disease burden as it relies on known cancer mutations allowing more accurate disease monitoring compared to cfDNA [10,11]. However, unlike solid tumors and adult cancers, the prognostic value of ctDNA to detect MRD in pediatric childhood ALL remains largely unknown [12,13,14,15]. This disparity can be partially attributed to the lack of basic studies using clinically relevant experimental models that can accurately recapitulate childhood ALL relapse [9,16,17].

To address these challenges, we employed an orthotopic patient-derived xenograft (PDX) mouse model to investigate ctDNA dynamics in infant MLL-r ALL. This model effectively replicates the development and progression of systemic disease, where leukemia cells engraft in the BM then circulate to the spleen, liver, and peripheral blood (PB), closely mimicking the disease in patients [18,19]. Therefore, this PDX model has been recognized as one of the best experimental systems to study childhood ALL. To simulate the clinical stages of ALL, we treated the mice with the menin inhibitor SNDX-50469, which we previously showed induced extended remissions of MLL-r ALL PDXs followed by highly reproducible disease relapse or cure in several mice [20]. This novel class of targeted therapy works by disrupting binding of the menin scaffold protein to KMT2A fusion oncoproteins or mutant *NPM1*, thereby downregulating the aberrant leukemogenic transcriptional program [21,22]. Using this experimental model offers an opportunity to quantify MRD in the BM and understand the correlation between ctDNA with treatment response and MRD burden. The work presented here provides important proof-of-principle data and further supports the application and feasibility of ctDNA analysis in preclinical studies.

## 2. Materials and Methods

### 2.1. Patient-Derived Xenografts (PDX) Cells

All PDXs (MLL-1, MLL-2, MLL-7) used originated from patient BM or PB biopsy samples and engrafted into non-obese diabetic/severe-combined immunodeficient/interleukin-2 receptor γ–negative (NOD.Cg-*Prkdc^scid^ Il2rg^tm1^*Wjl/SzJAusb, NSG) mice, as previously described [16]. The PDX cells were subsequently collected from the spleens of highly engrafted mice and purified by Ficoll density gradient separation, followed by cryopreservation. All studies were conducted under the approval of the Animal Care and Ethics Committee of UNSW Sydney (Sydney, Australia). The clinical characteristics of the original patient samples and ex vivo PDX features are described in Table 1. For in vivo studies, PDX cells were retrieved from cryostorage and washed with RPMI-1640 media with 5% fetal bovine serum (FBS) (ThermoFisher, Sydney, NSW, Australia), then resuspended in Quality Biologicals Serum Free-60 (QBSF-60) (Quality Biological, Gaithersburg, MD, USA). The cells were counted by trypan blue exclusion assay and transferred to 6-well, U-bottom tissue culture plates at previously optimized cell density. The PDX cells were equilibrated in an incubator at 37 °C and 5% CO_2_, for at least 3 h (h) before DNA extraction.

### 2.2. qPCR Assay Design for Detecting ctDNA

The fusion sequences of the MLL-r ALL *KMT2A* from the original diagnostic patient samples were kindly provided by the Molecular Diagnostic Group (Children’s Cancer Institute, Sydney, NSW, Australia). The breakpoint sequences were determined by the Meyer Lab at the Diagnostic Centre of Acute Leukemia (Frankfurt, Germany) as previously described [3]. Primer design for *KMT2A* fusion assays were performed using Primer3Plus and Primer Blast and the resulting primers were tested for optimal annealing temperatures. Primer sequences are listed below:

Forward Primer MLL-1 (5′-3′): CACAGGAGGATTGTGAAGCA

Reverse Primer MLL-1 (5′-3′): GGTGCTCCTGTTGGTTACCT

Forward Primer MLL-2 (5′-3′): GGAGAGCTTTGGTCAGTGTTG

Reverse Primer MLL-2 (5′-3′): TGCTGTGGTCTGAATGTGTC

Forward Primer MLL-7 (5′-3′): AAGGGATGCTATTGATGGTTATTTT

Reverse Primer MLL-7 (5′-3′): AACACCATGGCAAACCCATTC

To optimize each *KMT2A* fusion assay for ctDNA quantification, qPCR was performed using 10-fold serial dilutions of genomic DNA isolated from PDX cells, and data were interpreted as previously described [4]. The controls for all qPCR assays were genomic DNA isolated from ALL-19 PDX cells (*KMT2A* fusion-negative) and non-template controls (NTCs), which were used to set a threshold for each assay and minimize background noise, thereby avoiding false positive results.

### 2.3. Mouse Studies with MLL-r ALL PDXs

All animal experiments were conducted under approval by the Animal Care and Ethics Committee (UNSW Sydney). For engraftment studies, six female immune-deficient NSG mice were inoculated with 2–3 × 10^6^ MLL-r ALL PDX cells (MLL-1, MLL-2, or MLL-7) via tail vein injection to establish the PDX model. To ensure their well-being daily monitoring and weekly weighing of all mice were conducted. The % human CD45^+^ (% huCD45^+^) population in the PB of the mouse was used to measure engraftment on a weekly basis, as previously described [23]. Every week, around 150 µL of PB from the lateral tail vein was collected into Mini-Collect^®^ EDTA tubes. Up to 50 µL of PB was then used for determining the % huCD45^+^ and the remaining was processed into plasma within 4 h of bleeding mice, for subsequent ctDNA analysis.

The SNDX-50469 drug was kindly provided by Syndax (Syndax Pharmaceuticals, Waltham, MA, USA). In each PDX group (MLL-1, MLL-2 and MLL-7), when *n* = 10–12 of mice reached a median engraftment of ≥1% huCD45^+^, they were randomized into two groups (*n* = 5–6) and treated with either 120 mg/kg SNDX-50469 or vehicle (0.5% [*w*/*v*] methylcellulose) twice per day using oral gavage for 28 days. The efficacy of SNDX-50469 was determined by mouse EFS, which was defined as the number of days from treatment initiation until event defined as 25% huCD45^+^. This was calculated by interpolating between bleeds directly preceding and following events, assuming log-linear growth. Drug efficacy was also measured using stringent objective response measures (ORM), as described previously [24]. A group size of six mice/group was selected as it provides > 85% power at an alpha of 0.05 to detect a two-fold difference in EFS between SNDX-50469-treated mice and vehicle-treated mice.

### 2.4. Assessment of Leukemia Organ Filtration

Leukemia organ infiltration was measured on Day 0, 14, and 28 and when mice relapsed at 1% huCD45^+^ or endpoint following treatment initiation, whichever occurred first (an additional five or six mice per timepoint). For each timepoint, blood collected by cardiac puncture, spleen, and BM of the mice were collected. In our previous study of SNDX-50469 efficacy, mice engrafted with the MLL-7 PDX relapsed before Day 84, those with the MLL-2 PDX relapsed at around Day 84, and those with the MLL-1 PDX remained in remission well beyond Day 84 [20]. Therefore, the Day 84 endpoint was chosen to investigate ctDNA dynamics at these stages of disease remission and relapse. Mice were euthanized by CO_2_ asphyxiation away from the other mice. Spleens and approximately 1 mL of blood were collected via cardiac puncture to assess leukemic infiltration by flow cytometry. For MRD analysis, DNA was extracted from BM samples using Nucleobond Kits (Scientifix, Clayton, VIC, Australia), as per the manufacturer’s instructions. DNA was quantified using a Nanodrop ND-1000 spectrophotometer. MRD detection was performed and analyzed in compliance with the EuroMRD guidelines as described previously [4]. DNA from naive mice was used as a negative control.

### 2.5. cfDNA Isolation and Quantification

The remaining volume from weekly PB samples and blood collected by cardiac puncture in Mini-Collect^®^ EDTA tubes were fractionated into plasma by centrifugation at 2500× *g* for 10 min at 4 °C. To remove remaining cells, the supernatant was then transferred into a 1.5 mL Eppendorf tube and centrifuged at 3500× *g* for 10 min at 4 °C. The supernatant was subsequently transferred to another Eppendorf tube for storage at −80 °C. Total cfDNA was extracted using Norgen Plasma/Serum Cell-Free Circulating DNA Purification Mini Kits according to the manufacturer’s protocols. Total cfDNA concentration and purity were measured using TapeStation Analysis version 3.2 (Agilent, Chatswood, NSW, Australia). Extracted cfDNA samples were stored at −80 °C until use.

### 2.6. Quantification of ctDNA Using Droplet Digital PCR (ddPCR)

The qPCR primers for *KMT2A* fusion amplification were used for ddPCR assays for absolute quantification of ctDNA. A maximum of 20 μL of ddPCR reaction mix was prepared consisting of 12.5 μL of 2× ddPCR Master Mix for probes (Bio-Rad, Gladesville, NSW, Australia), 1.25 μL of 20× primers (500 nmol/L), and 0.5 ng of DNA and nuclease-free water. Droplets were generated by the QX200 Droplet Generator followed by DNA amplification using a C1000 Touch Thermal Cycler (Bio-Rad). Droplet counting and thermal cycling conditions were conducted as previously described [25]. Reactions with <10,000 droplets were excluded from the analysis. The normalization of raw DNA copy numbers (copies/mL) was calculated using the formula below, as described in [26]:$$\frac{Events\times 20}{Input\;cfDNA\;volume\;(µL)}\times \frac{Elution\;volume\;(µL)}{Plasma\;volume\;(µL)}$$

For this equation, input cfDNA volume indicates the volume added to the ddPCR reaction, plasma volume is the initial volume after subsequent plasma processing, and elution volume is the volume of eluted DNA following cfDNA extraction. The threshold for positive target detection was set above two or more positive droplets (≥10 mutant copies/mL of plasma) to ensure ddPCR assay specificity [26].

### 2.7. Statistical Analysis

To compare total cfDNA and ctDNA, Spearman’s correlation and unpaired student *t* test with Welch’s correction were used. Poisson statistics were used for random distribution analyses to compare ctDNA concentrations between test and control samples using ddPCR. EFS curves were generated using Gehan–Breslow–Wilcoxon test. All data analyses were performed using GraphPad Prism 10 software (GraphPad 9.0, La Jolla, CA, USA). A *p*-value of < 0.05 was considered statistically significant.

## 3. Results

### 3.1. Analysis of Patient-Specific ctDNA Allows for Accurate Preclinical Monitoring of Disease Progression

We investigated whether the patient-specific ctDNA assays could be used for longitudinal monitoring of disease progression and treatment response in a PDX mouse model of pediatric MLL-r ALL. Figure 1A represents an overview of the preclinical model. NSG mice were inoculated with MLL-1, MLL-2, or MLL-7 PDX cells and disease burden was assessed weekly by flow cytometry analysis of the % huCD45^+^ as well as cfDNA and ctDNA detection using Tapestation and ddPCR, respectively. Human cancer DNA can be easily discriminated from host mouse DNA in PDX models by targeting human cancer-specific sequences such as chromosomal translocations through qPCR [27]. Each qPCR assay conformed to the standardized EuroMRD guidelines (Figure 1B, Appendix A) [4].

We observed a significant increase in total cfDNA levels in all mice engrafted with MLL-r ALL PDXs at three weeks post inoculation compared to naive mice (Figure 2A). Interestingly, there was a large difference between the levels of total cfDNA measured from the MLL-1 cohort (maximum 600 pg/µL) and MLL-2 (maximum 7000 pg/µL) and MLL-7 (maximum 6000 pg/µL), indicating variability between PDXs of the same ALL subtype (Figure 2A). We then measured the levels of ctDNA (copies/µL) and observed a significant correlation with leukemia burden (% huCD45^+^) in mice engrafted with MLL-1, MLL-2, and MLL-7 PDXs (r = 0.75, 0.88, 0.85 *p* < 0.0001, respectively) (Figure 2B–D). Of note, MLL-2 exhibited the highest levels of ctDNA (Figure 2C), consistent with it eliciting the highest levels of cfDNA (Figure 2A). We also observed a significant correlation between cfDNA levels and % huCD45^+^ for MLL-2 and MLL-7 but not MLL-1 at one–three weeks post inoculation (Appendix A). Despite the limited sample size (*n* = 3 PDXs), a general trend was observed wherein the PDX with the lowest cfDNA concentration, MLL-1, also exhibited the lowest ctDNA levels, and similarly MLL-2 demonstrated both the highest cfDNA and ctDNA concentrations (Figure 2B,C and Appendix A). These findings highlight that in preclinical models, ctDNA allows for accurate longitudinal measurement of disease progression.

### 3.2. Modeling Remission and Relapse in an Orthotopic Mouse Model of MLL-r ALL PDXs Treated with SNDX-50469

We next aimed to mimic the clinical features of pediatric patients with MLL-r ALL treated with small-molecule menin inhibitors, which have shown exceptional efficacy in preclinical models, supporting their progress to clinical trials [20,21,22]. NSG mice were inoculated with MLL-1, MLL-2, or MLL-7 PDXs, monitored for engraftment, followed by treatment with SNDX-50469 as described above. By Day 21 after treatment initiation, all SNDX-50469-treated mice achieved remission (0% huCD45^+^ in the PB). In contrast, all mice treated with vehicle control reached event within 28 days (Figure 3, top panel). Mice engrafted with MLL-1 and MLL-2 PDXs (four of five mice) experienced an excellent prolongation in EFS following the 28-day SNDX-50469 treatment regimen (Figure 3A,B, lower panels). Extended survival was also observed in treated mice in the MLL-7 cohort, albeit for a shorter period (Figure 3C, lower panel). The EFS of SNDX-50469-treated mice compared to vehicle treated mice was significantly extended with treated minus control (T-C) values of >63.6 days (*p*-value 0.0039) for MLL-1, > 83 days (*p*-value 0.0013) for MLL-2, and 64 days (*p*-value 0.0013) for MLL-7 (Figure 3 and Appendix A). In summary, SNDX-50469 administered as a single agent for 28 days markedly improved the EFS of mice engrafted with aggressive MLL-r ALL PDXs. As the MLL-r ALL PDX models demonstrated different degrees of response to SNDX-50469 (mice engrafted with MLL-1 and MLL-2 PDXs showed extended remissions while mice engrafted with MLL-7 relapse at <70 days), this enabled assessment of ctDNA as a biomarker for treatment response and disease progression across different clinical scenarios.

### 3.3. ctDNA Is Detectable in Mice with Low Leukemia Burden in PB During and Following SNDX-50469 Treatment

Changes in ctDNA dynamics have been shown to allow for longitudinal monitoring and drug treatment [28,29]. However, it is unclear whether similar predictive measures could be used to assess drug response in preclinical models of infant MLL-r ALL. We asked whether the changes in ctDNA or cfDNA during and following treatment with SNDX-50469 were predictive of disease outcome in our PDX cohort. We analyzed MLL-r ALL growth kinetics by weekly measurements of % huCD45^+^ by flow cytometry as well as ctDNA and cfDNA over a time course of 0 to 84 days during and post treatment of mice engrafted with MLL-r ALL PDXs with SNDX-50469.

The % huCD45^+^ in the PB of all the mice in the MLL-1 cohort began to decline immediately after the initiation of SNDX-50469 treatment, was barely detectable after 28 days (<0.1% huCD45^+^) and remained so until the end of the study on Day 84 (Figure 4A, left panel). Interestingly, there was an increase in the cfDNA levels throughout the observation period (Figure 4A, middle panel, Appendix A, left panel). The analysis also showed an initial increase in ctDNA (Day 0–28; Figure 4A, right panel; Appendix A, right panel), which we reasoned could either be predictive of cell death due to drug response or indicate the presence of MRD at these time points. The decrease in ctDNA levels at Day 84 compared to baseline pretreatment levels for MLL-1 was statistically significant (Appendix A).

Four out of five mice engrafted with MLL-2 also reached engraftment levels of <0.1% huCD45^+^ for 84 days (Figure 4B, left panel). While a variety of trends was observed in the cfDNA levels, the ctDNA levels remained approximately equivalent (Figure 4B, middle and right panels, respectively; Appendix A). In the case of MLL-7 engrafted mice, while they reached <0.1% huCD45^+^ for a few weeks after initiation of SNDX-50469 treatment, the disease began to re-emerge after Day 49 (huCD45^+^ > 1%) (Figure 4C, left panel), with cfDNA and ctDNA levels fluctuating during the observation period (Figure 4C, middle and right panels, respectively; Appendix A). There were no significant differences between the ctDNA levels from baseline to endpoint in MLL-2 and MLL-7 (Appendix A, respectively).

In contrast to our findings above during establishment of disease in mice (Figure 2), we observed no trend between the % huCD45^+^ in the PB and ctDNA levels following drug treatment (Figure 4, Appendix A). To address this discrepancy and gain a greater understanding of disease dynamics, our subsequent experiments focused on exploring organ infiltration patterns of MLL-r ALL and comparing these findings with ctDNA levels. This approach allowed us to investigate the potential relationship between tissue-specific leukemia burden and the presence of ctDNA, providing potential insights into the utility of ctDNA as a biomarker for monitoring treatment response and detecting occult disease in infant MLL-r ALL.

### 3.4. ctDNA Levels Reflect MRD Burden in Mice Engrafted with MLL-r ALL PDXs

The capacity to detect ctDNA at low levels as a potential predictor of MRD in preclinical models of infant MLL-r ALL remains unclear. To examine the extent to which ctDNA reflects MLL-r ALL burden in MRD sites, plasma ctDNA concentrations were compared to BM MRD as well as the % huCD45^+^ cells in the spleen and in cardiac puncture blood collected from euthanized mice at Day 0, 14, and 28 and at 1% huCD45^+^ or endpoint (Figure 5). For MLL-1, the presence of leukemia cells in cardiac puncture blood as well as the spleen samples was at almost undetectable levels upon treatment initiation (Figure 5A, left and second from left panels, respectively). However, the MLL-1 mice were MRD positive in the BM on Day 0–14, followed by a significant decline and negative MRD levels in subsequent timepoints (Figure 5A, second from right panel). Interestingly, this was reflected in the ctDNA levels detected in the plasma samples of these mice at these time points, i.e., an increase in the ctDNA levels between Day 0–14 followed by a significant decrease compared to the baseline levels at Day 84 (Figure 5A, right panel and Figure 4A, right panel).

Conversely, for MLL-2, although there was a maintained decrease in the % huCD45^+^ in cardiac puncture and spleen samples, except for one mouse at Day 84 (Figure 5B, left and second from left panels, respectively), the mice were MRD positive in the BM at all timepoints (Figure 5B, second from right panel). This was consistent with the increase in ctDNA levels observed in all time points (Figure 5B, right panel and Figure 4B, right panel). In the MLL-7 cohort, the increase in leukemia cells in blood collected by cardiac puncture indicated relapse on Day 56 (Figure 5C, left panel). This was reflected in the presence of leukemia in the spleen, as well as MRD positivity at all time points (Figure 5C, two middle panels). Although the ctDNA levels for these timepoints were not representative of these sites (Figure 5C, right panel and Figure 4C, right panel), there was no significant difference between ctDNA levels on Day 56 and Day 0 (Appendix A), suggesting MRD positivity and the relapse observed at Day 56. These findings underscore the ability of ctDNA to reflect changes in disease across different tissue compartments and predict MRD, despite negligible levels of leukemia in the PB.

## 4. Discussion

Although there are promising studies on the prognostic value of ctDNA in solid tumors and adult cancers, our understanding of the predictive value of ctDNA in childhood ALL remains limited [12,30]. In this study, we demonstrated the utility of ctDNA as a potential biomarker for monitoring MRD and treatment response in a preclinical PDX model of pediatric MLL-r ALL. The detection of ctDNA in mice with low leukemia burden and using low blood volumes enabled the longitudinal monitoring of disease progression. Moreover, ctDNA levels showed strong potential as an indicator of disease status, with a notable correlation to MRD positivity across most time points and MLL subtypes. The detection of ctDNA in the scenario of negative MRD may indicate the presence of MRD with higher sensitivity that may not be observed with conventional testing.

There is an urgent need for basic research to realize the clinical potential of ctDNA from bench to bedside [31]. However, to date, there are limited published reports focusing on the use of animal models for studying the kinetics of ctDNA during different disease stages [29,32,33]. In these studies, ctDNA was reported to measure disease burden, monitor targeted treatment response and predict disease outcome with high sensitivity. To the best of our knowledge, this is the first study on the predictive value of ctDNA using a PDX mouse model of pediatric ALL which maintains the genetic and phenotypic characteristics of the original cancer, thus allowing for controlled and longitudinal monitoring of ctDNA levels in relation to disease progression and treatment response [18,19].

The utilization of the menin inhibitor SNDX-50469 in MLL-r ALL PDX models yielded diverse treatment responses, ranging from prolonged remission to relapse. This heterogeneity in treatment outcomes provided a unique opportunity to investigate the relationship between ctDNA dynamics and MRD burden across different response groups. Crucially, PDX models enable the correlation of ctDNA levels with leukemia burden in specific organs, such as the spleen and BM, providing insights that are often unattainable in clinical settings due to practical and ethical constraints. This organ-specific information, combined with the ability to standardize experimental conditions and test novel treatment strategies, enhances our understanding of ctDNA as a surrogate biomarker in ALL and accelerates the development of personalized medicine approaches.

Our results showed that cfDNA and ctDNA can be detected in small volumes of plasma from mice with MLL-r ALL, even when the leukemic burden in the PB was close to zero. This aligns with previous studies and highlights that ctDNA might improve MRD detection in certain clinical settings, including low disease burden stages when ctDNA levels are typically very low [29]. We also observed a significant variability in the ctDNA and cfDNA levels across different PDXs. However, despite the limited sample size of three PDXs, MLL-1 and MLL-2 demonstrated the lowest and highest cfDNA and ctDNA concentrations, respectively. This trend is consistent with a recent report of strong correlation between ctDNA % and cfDNA concentration [34]. Although the evidence for the differences in ctDNA shedding between different tumors is still limited, it has been shown that some cancers can yield low ctDNA amounts, despite being at metastatic or refractory stages [35]. This variability could be attributed to the potential presence of MRD in various organs, which may not be detectable in PB. These hidden reservoirs of leukemia cells could continue to shed DNA into the circulation, resulting in detectable ctDNA levels even when leukemia cells are not apparent in the blood, as shown in chronic lymphocytic leukemia and other cancers [36,37]. This variability persisted despite maintaining consistent pre-analytical and inter-assay conditions, including standardized blood processing protocols. The discrepancy may also be due to tumor heterogeneity, differences in engraftment efficiency, and individual host factors influencing disease progression and tumor cell turnover rates [38]. Further, the high variability of cfDNA and ctDNA yields between individual mice in each of the MLL-r ALL PDXs is supported by previous reports of patients with seemingly similar disease classifications exhibiting varied responses and outcomes [39,40]. These biological differences highlight the complexities of ctDNA as a potential biomarker and the need for comprehensive, personalized, and multi-faceted approaches to disease monitoring and treatment strategies in MLL-rearranged leukemias.

While there was a significant linear relationship between % huCD45^+^ in the PB and ctDNA and cfDNA levels in plasma, this concordance was lost after drug treatment, again suggesting that cfDNA and ctDNA may be a result of shedding not only from circulating cells in PB, but organs known to harbor leukemia cells during therapy [36]. In current PDX models of ALL, treatment response is assessed by the enumeration of % huCD45^+^ in mouse PB as a surrogate indicator of disease burden [18]. While effective, this technique cannot assess the persistence of MRD and the effect of treatment on these relapse inducing cells. Also, BM biopsy cannot inform about relapses occurring in extramedullary sites including the spleen. While MRD is the measurement of residual cancer cells in a single site, our results show that ctDNA levels represent the average of the total body disease burden, and thus, could have significant prognostic value in determining overall treatment response and MRD.

Despite the advantages described above, several limitations of the study should be noted. First, an analysis of additional timepoints after Day 84 could have provided more information on ctDNA dynamics compared to the conventional measures of disease burden. The positive MRD at Day 84 in the MLL-2 cohort suggests that the ctDNA could have increased after this timepoint. Further studies should be conducted to investigate the parallel between ctDNA in other extramedullary tissues including the lymph nodes, liver, and central nervous system, as leukemia is known to disseminate in these sites [41]. Given the differences observed between MLL-r ALL PDXs, particularly the variability of ctDNA levels in the MLL-7 cohort, it will be of significant interest to interrogate ctDNA in other ALL PDXs. Despite these limitations, our findings support the introduction of ctDNA analysis as a complementary approach to consensus methodologies of assessing MRD.

## 5. Conclusions

In summary, here, we provide the first report describing the detection of ctDNA in preclinical models of MLL-r ALL and demonstrate the potential of this strategy for molecular disease monitoring in known sites of leukemic infiltration. Our findings serve as a foundation for future studies that will include prospective ctDNA analysis using a range of high-risk ALL biomarkers for a comprehensive assessment of disease burden.

## Figures and Tables

**Figure 1 cancers-16-03990-f001:**
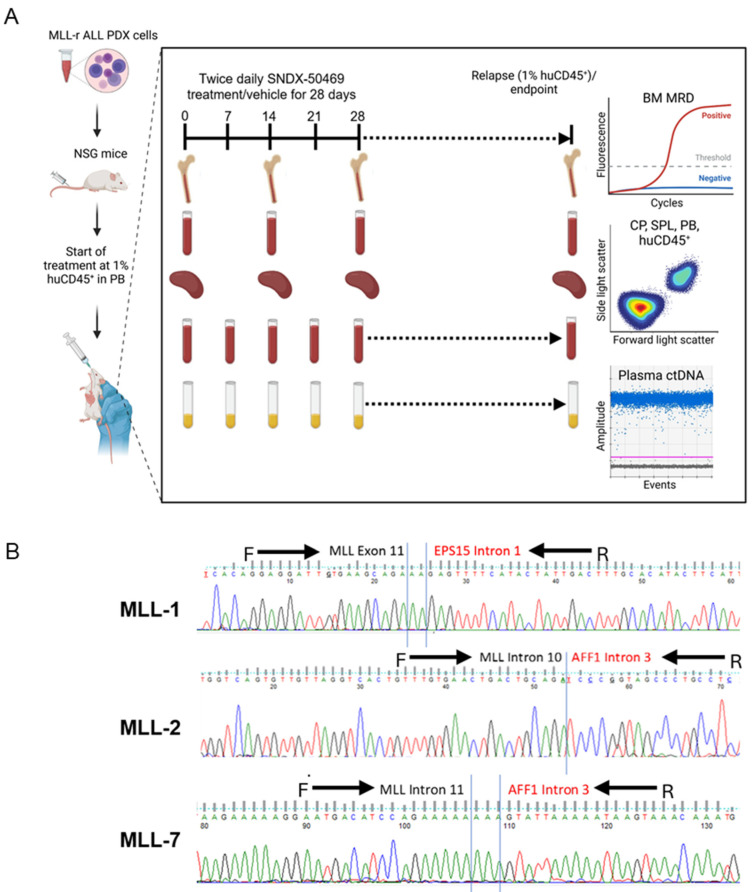
Experimental model of ctDNA analysis in mice engrafted with MLL-r ALL. (**A**) Schematic of workflow for the preclinical model of NSG mice engrafted with MLL-r ALL PDXs and treated with SNDX-50469. Treatment commenced when mice reached 1% huCD45^+^. SNDX-50469 or vehicle control were administered via oral gavage (*n* = 10–12). At weekly timepoints until 1% huCD45^+^ relapse/endpoint, peripheral blood (PB) was collected via tail vein bleeds for flow cytometry analysis of % huCD45^+^, and plasma was analyzed for ctDNA concentration using ddPCR. On Day 0, 14, and 28 and the 1% huCD45^+^ relapse/endpoint, cardiac puncture (CP) and spleen (SPL) samples were analyzed for % huCD45^+^ and bone marrow (BM) samples were analyzed for MRD quantification (*n* = 5 or 6 per timepoint per PDX). (**B**) Sequencing alignment of *KMT2A* fusions for MLL-1, MLL-2, and MLL-7 PDXs. Black arrows indicate the forward and reverse primers that cover the *KMT2A* and fusion partner regions for fusion breakpoint amplification via qPCR and ddPCR.

**Figure 2 cancers-16-03990-f002:**
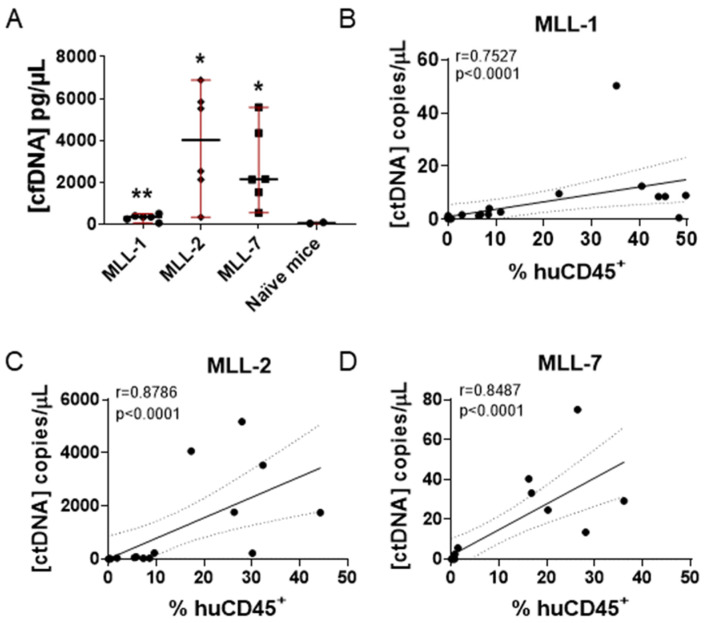
Comparison of cfDNA and ctDNA concentrations and % huCD45^+^ in NSG mice engrafted with MLL-r ALL PDXs. (**A**) Comparison of cfDNA values obtained from two naive mice with PDX engrafted mice (six individual mice) at three weeks post inoculation (unpaired *t* test with Welch’s correction). ** = *p* < 0.01, * = *p* < 0.05. Median ± range are indicated by the black horizontal line and red bars, respectively. (**B**–**D**) Spearman correlation analysis of ctDNA concentrations and % huCD45^+^ measured at one-three weeks post inoculation in six mice engrafted with (**B**) MLL-1 (*n* = 18) (**C**) MLL-2 (*n* = 18), and (**D**) MLL-7 (*n* = 18) PDXs. Dashed lines represent the 95% confidence limits of the best fit line. r = Spearman correlation coefficient; *p* = Spearman two-tailed correlation test.

**Figure 3 cancers-16-03990-f003:**
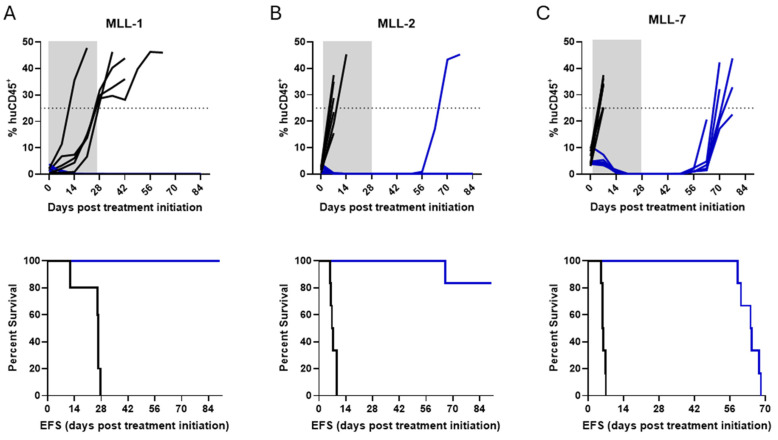
In vivo efficacy of SNDX-50469 against MLL-r ALL PDXs. Top panels, leukemia engraftment measured by PB % huCD45^+^ plotted over time; lower panels, Kaplan–Meier curves demonstrating the percent survival of mice engrafted with (**A**) MLL-1, (**B**) MLL-2, and (**C**) MLL-7 PDX cells (11–12 mice per PDX) and treated with SNDX-50469 or vehicle control (twice daily treatment for 28 days). The dashed line across 25% huCD45^+^ represents event. Gray area represents the treatment duration. Vehicle control (black lines) and SNDX-50469 (blue lines).

**Figure 4 cancers-16-03990-f004:**
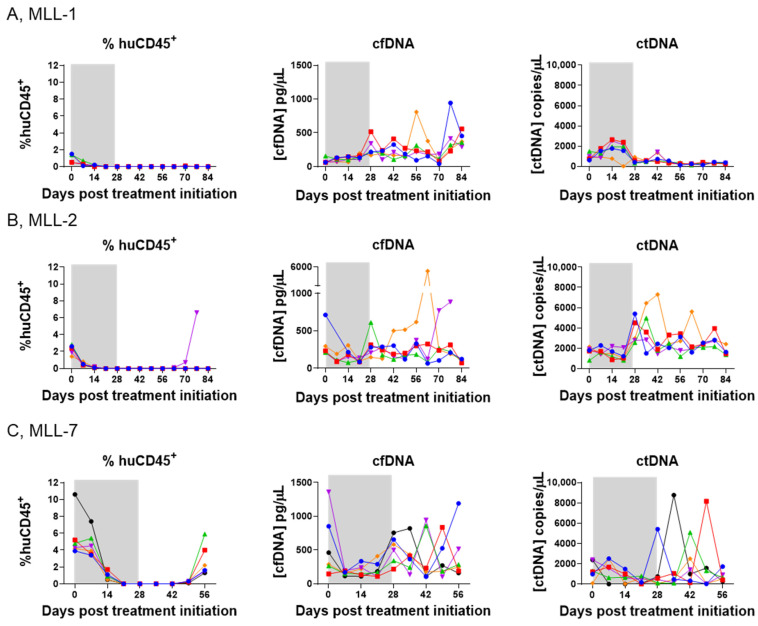
Comparison of disease burden in the PB with plasma cfDNA and ctDNA concentrations in SNDX-50469-treated NSG mice engrafted with MLL-r ALL PDXs. Left panels, flow cytometry analysis of % huCD45^+^ cells in the peripheral blood; middle panels, TapeStation analysis of cfDNA; and right panels, ddPCR analysis of ctDNA at weekly timepoints from Day 0 (SNDX-50469 treatment initiation) in tail vein blood samples from 5–6 mice inoculated with (**A**) MLL-1, (**B**) MLL-2, and (**C**) MLL-7 PDXs. Each color represents an individual mouse. The shaded grey area indicates the treatment window.

**Figure 5 cancers-16-03990-f005:**
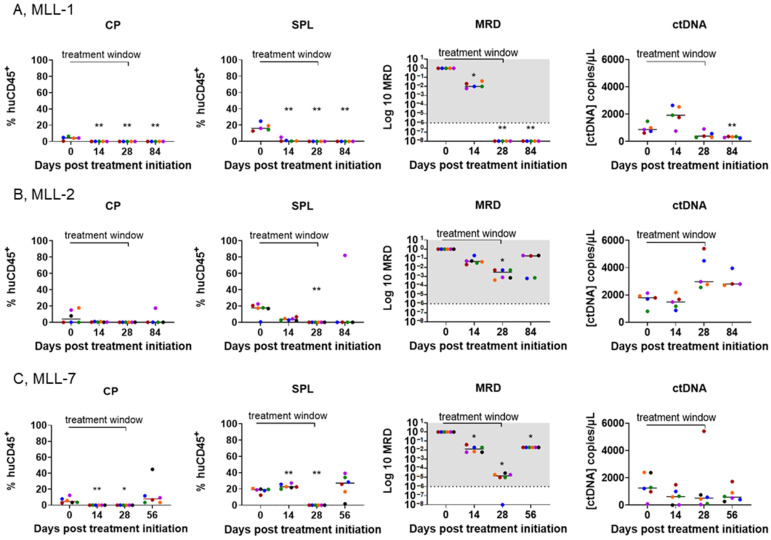
ctDNA levels reflect multiple compartments of disease burden in mice engrafted with MLL-r ALL PDXs and treated with SNDX-50469. Comparison of disease burden in the cardiac puncture (CP, left panels), spleen (SPL, second from left panels), BM MRD (second from right panels), and plasma ctDNA concentration (right panels) from the tail vein blood of SNDX-50469-treated NSG mice engrafted with (**A**) MLL-1, (**B**) MLL-2, and (**C**) MLL-7 PDXs. Each color represents one mouse. Dashed lines represent the median value. The shaded gray area indicates the range of MRD positivity. Analysis of ctDNA was conducted by ddPCR and analysis of the % huCD45+ cells in the cardiac puncture and spleen performed by flow cytometry at timepoints from Day 0 (SNDX-50469 treatment initiation) in 5–6 mice. One mouse was excluded from each timepoint group of the MLL-1 cohort and one mouse was excluded from the Day 84 group of mice inoculated with MLL-2 PDX due to unexpected death. Each timepoint was compared to the respective Day 0 post treatment initiation (unpaired *t* test with Welch’s correction). ** = *p* < 0.01, * = *p* < 0.05.

**Table 1 cancers-16-03990-t001:** Clinical, demographic, and molecular characteristics of the PDXs used in this study. These characteristics were previously described [19]. MLL-r, mixed-lineage leukemia (*KMT2A*) rearranged.

PDX	Age at Diagnosis (y)	Lineage	Ethnicity	Stage of Disease	Chromosomal Translocation	Structural Variants (SVs)
MLL-1	<1	B-ALL	Mixed or unknown	Diagnosis	t(1;11)	*KMT2A::EPS15*
MLL-2	<1	B-ALL	European	Diagnosis	t(4;11)	*KMT2A::AFF1; DUX4 SV*
MLL-7	<1	B-ALL	Mixed or unknown	Diagnosis	t(4;11)	*KMT2A::AFF1; MEF2D::PTMA; KMT2A::PTPRC*

## Data Availability

All data generated or analyzed during this study are included in this manuscript. Appendix A is available.

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
