# Peer review of "Patient-Specific Circulating Tumor DNA for Monitoring Response to Menin Inhibitor Treatment in Preclinical Models of Infant Leukemia"

_cancers, 2024, doi:10.3390/cancers16233990_

Round 1
Reviewer 1 Report
Comments and Suggestions for Authors
The authors analyze circulating tumor DNA (ctDNA) as a disease monitoring tool for acute lymphoblastic leukemia (ALL). They utilize a PDX mouse model for analysis and report that this is the first analysis looking at ctDNA with ALL in a mouse model, so I do think it is significant and novel research. Some of the set-up and analysis should be clearer. For that reason, I would recommend to accept with minor revisions as outlined in detail below.
Introduction - cell-free DNA and its use in disease monitoring (compared to ctDNA) is not explained thoroughly despite being a main component of analysis in the results.
Methods
I am a little confused with the number of mice used. For example, it says a group of 6 mice/group were selected for each cell line to start in section 2.3 line 131 but then in lines 143-144 its n=6 per group of treatment vs vehicle which would mean originally 12 were inoculated per cell line. But then figure 2 says n=18 per MLL-r type for the ctDNA vs %huCD45+ data (even though it does not appear to be 18 points on each graph unless there are ones at the same exact spot). Are these separate studies? The appropriate number of mice needs to be clear throughout.
In 2.3 it says PB measurement was through lateral tail vein collection 136-139 and that was used for ctDNA analysis. But then later cardiac puncture is used for the 0, 14, 28 days. I believe the 0, 14, 28 were an endpoint collection and why there is a different method, but it is not clear how many were sacrificed at each of those time points vs how many started in the group; were those mice separate from the treatment vs vehicle group study? This is unclear. And is the lateral vein and cardiac puncture blood data used in the same analyses? It would be best to compare this information within the same collection method. For example, is the data in figure 4 only from tail vein or does that include cardiac puncture data points too?
2.5 lines 167-170: was it plasma or serum? No anticoagulant is mentioned to be added.
Results
Figure 1 was helpful to understand but some things could be added to clarify. The number of mice should be mentioned somewhere in the diagram including how many were euthanized for the terminal collection points. The type of blood collection (cardiac puncture vs tail vein) can be included for each blood sampling.
Figure 2 – For A., it is mentioned to be at time point 3 weeks post inoculation. Is it the same for B-D. Can supplementary figure s2 information be moved to be included in the main Figure 2?
Figure 4 – It could be included in the caption that each color represents one mouse each. There is a lot of variability between mice in cfDNA and ctDNA especially in the MLL-2 and 7 making it difficult to see a pattern. In a supplement, can the values be shown as an average with error bars to see if a pattern can be better visualized?
Section 3.3 lines 278-280 “cfDNA levels generally decreased throughout the treatment” that does not seem to be reflected in figure 4B middle; there is a variety of trends among the mice in that group.
Figure 5 – the statistics must be included among days for each panel to demonstrate if the increases or decreases are statistically significant in each analysis. The points in each graph can be color coded to match the same mouse among each assessment. This could show when there is variability. For example, for the CP outlier for MLL-7 in the left panel does that associate with the high data point outlier in ctDNA at day 28 in the right panel?
Discussion
Lines 333-335 “ctDNA levels accurately predicted the presence of MRD”. This is based off figure 5, and I do not know if I have that takeaway definitively from the data shown there. The ctDNA does not necessarily follow the same pattern as the MRD, and is there a sort of range of positivity for ctDNA copies/ul? If it is anything above 0, then this statement would be inaccurate for MLL-1 days 28 and 84.
The graphs show quite a bit of variability for the ctDNA among mice within a MLL type. This should be addressed in the discussion. Does this complicate the translatability?
In the results there is analysis of the cell-free DNA but in the discussion, there is little mention of it or its comparison to the ctDNA findings.
Author Response
Introduction - cell-free DNA and its use in disease monitoring (compared to ctDNA) is not explained thoroughly despite being a main component of analysis in the results.
Response: We thank the Reviewer for this comment and have expanded the Introduction section to address this (lines 68-75).
I am a little confused with the number of mice used. For example, it says a group of 6 mice/group were selected for each cell line to start in section 2.3 line 131 but then in lines 143-144 its n=6 per group of treatment vs vehicle which would mean originally 12 were inoculated per cell line. But then figure 2 says n=18 per MLL-r type for the ctDNA vs %huCD45+ data (even though it does not appear to be 18 points on each graph unless there are ones at the same exact spot). Are these separate studies? The appropriate number of mice needs to be clear throughout.
Response: We have modified the text to better explain the number of mice used in this study. The experiment used in Figure 2 was conducted separately from the treatment studies described later in the manuscript. In this study, 6 mice were inoculated with each PDX type (MLL-1, MLL-2, and MLL-7). Engraftment was measured at three time points: 1, 2 and 3 weeks post-inoculation. This resulted in 18 data points per MLL-r type (6 mice × 3 time points) as shown in the correlation graphs in Figure 2B-D. The study mentioned in lines 149-152 is a separate experiment; 6 mice per group were used for each treatment condition (treatment vs. vehicle). This resulted in a total of 12 mice per cell PDX (6 for treatment + 6 for vehicle). To improve clarity, we have updated the text accordingly to clearly differentiate between these separate studies and provide explicit details on the number of mice used in each experiment (lines 139;149-152; 226; 257).
In 2.3 it says PB measurement was through lateral tail vein collection 136-139 and that was used for ctDNA analysis. But then later cardiac puncture is used for the 0, 14, 28 days. I believe the 0, 14, 28 were an endpoint collection and why there is a different method, but it is not clear how many were sacrificed at each of those time points vs how many started in the group; were those mice separate from the treatment vs vehicle group study? This is unclear. And is the lateral vein and cardiac puncture blood data used in the same analyses? It would be best to compare this information within the same collection method. For example, is the data in figure 4 only from tail vein or does that include cardiac puncture data points too?
Response: We thank the reviewer for their question. Regarding our blood collection methods, lateral tail vein collection was used for ctDNA analysis throughout the study. However, cardiac puncture was performed at endpoint collections (days 0, 14, 28, and at relapse or day 84) for %huCD45+ analysis. For the treatment vs. vehicle study, 5-6 mice per group were used (10-12 total per PDX). Separate cohorts of 5-6 mice each were euthanized at days 0, 14, 28, and at relapse (1% huCD45+) or day 84 endpoint for organ infiltration assessment. Figure 4 contains data exclusively from plasma obtained by tail vein bleeds. The caption of Figure 4 has been revised to reflect this.
To improve clarity, we have updated the text accordingly to explicitly state the number of mice used in each group and the timeline for sample collection (lines 161-164). We have also indicated which data correspond to each collection method (captions for Figures 1, 4 and 5). We believe these clarifications will address the reviewer's concerns and enhance the reproducibility of our study.
2.5 lines 167-170: was it plasma or serum? No anticoagulant is mentioned to be added.
Response: Plasma was used for all analysis. The text was updated accordingly to mention EDTA tubes for anticoagulation (line178).
Figure 1 was helpful to understand but some things could be added to clarify. The number of mice should be mentioned somewhere in the diagram including how many were euthanized for the terminal collection points. The type of blood collection (cardiac puncture vs tail vein) can be included for each blood sampling.
Response: We thank the Reviewer for their suggestion to clarify the mouse numbers and method of collection in Figure 1. To strengthen the Figure, we have revised the Figure 1 caption accordingly.
Figure 2 – For A., it is mentioned to be at time point 3 weeks post inoculation. Is it the same for B-D. Can supplementary figure s2 information be moved to be included in the main Figure 2?
Response: In Figure 2B-D, the ctDNA values and %huCD45+ values were assessed in samples collected at 1-3 weeks post inoculation. Regarding Supplementary Figure S2, we understand the reviewer's suggestion to include Supplementary Figure S2 in the main Figure 2. However, we believe that keeping it as a supplementary figure is more appropriate since it provides supporting information specifically for Figure 2A, focusing on the similarity in relative cfDNA and ctDNA concentrations in MLL-1 and MLL-2. The information in Supplementary Figure S2, while valuable, is not central to the main conclusions drawn from Figure 2. We acknowledge that expanding the analyses shown in Supplementary Figure S2 to other PDX subtypes could provide additional insights. This is an avenue we are considering for future studies to further elucidate the trends observed in cfDNA and ctDNA concentrations across different MLL-r subtypes.
To improve clarity, we have updated the figure legend for Figure 2 to explicitly state the time points used for each panel (A-D). Additionally, we ensure that the connection between Figure 2A and Supplementary Figure S2 is clearly explained in both the main text (lines 248-249) and the supplementary figure legend.
Figure 4 – It could be included in the caption that each color represents one mouse each. There is a lot of variability between mice in cfDNA and ctDNA especially in the MLL-2 and 7 making it difficult to see a pattern. In a supplement, can the values be shown as an average with error bars to see if a pattern can be better visualized?
Response: We appreciate the reviewer's suggestions regarding Figure 4. As suggested, we have updated the figure caption to clearly state that each color represents an individual mouse. This clarification will help readers better understand the data presentation and the variability between individual animals. We acknowledge the high variability observed, particularly in MLL-2 and MLL-7 models. This variability reflects the biological heterogeneity between PDXs and individual responses to treatment. We chose to present individual mouse data to transparently show the range of responses and avoid masking important individual variations that might be lost in averaged data.
However, we understand the reviewer's point about difficulty in discerning patterns. Therefore, as suggested, we have added a supplementary figure showing the average values with error bars for each time point and PDX model (Supplementary Figure S4). This will provide an additional perspective on the overall trends that are not immediately apparent in the individual mouse data. In future studies, we aim to further investigate the factors contributing to these differences in patient plasma samples.
Section 3.3 lines 278-280 “cfDNA levels generally decreased throughout the treatment” that does not seem to be reflected in figure 4B middle; there is a variety of trends among the mice in that group.
Response: We thank the Reviewer for this suggestion. The text in section 3.3 was updated to address this (lines 312-313).
Figure 5 – the statistics must be included among days for each panel to demonstrate if the increases or decreases are statistically significant in each analysis. The points in each graph can be color coded to match the same mouse among each assessment. This could show when there is variability. For example, for the CP outlier for MLL-7 in the left panel does that associate with the high data point outlier in ctDNA at day 28 in the right panel?
Response: We appreciate the Reviewer’s suggestions and have updated Figure 5 accordingly to show the statistically significant differences between all timepoints compared to their respective Day 0 timepoint, as well as the variability of ctDNA concentrations between individual mice of the same PDX. We aimed to demonstrate the results in a fashion that highlights the differences between the various sites of leukemia.
Lines 333-335 “ctDNA levels accurately predicted the presence of MRD”. This is based off figure 5, and I do not know if I have that takeaway definitively from the data shown there. The ctDNA does not necessarily follow the same pattern as the MRD, and is there a sort of range of positivity for ctDNA copies/ul? If it is anything above 0, then this statement would be inaccurate for MLL-1 days 28 and 84.
Response: We appreciate the reviewer's comment regarding our statement on lines 333-335. We agree that the relationship between ctDNA levels and MRD is more nuanced than our original statement suggested. Regarding MLL-1 and the detection of ctDNA in the absence of MRD in spleen and bone marrow, this could indicate the presence of MRD below the detection limit by standard testing. This observation aligns with emerging research suggesting that ctDNA may offer enhanced sensitivity in detecting minimal disease burden. We have revised our statement to more accurately reflect the relationship observed between ctDNA levels and MRD presence (lines 386-389).
The graphs show quite a bit of variability for the ctDNA among mice within a MLL type. This should be addressed in the discussion. Does this complicate the translatability?
Response: The considerable variability in ctDNA levels among mice within each MLL subtype likely stems from multiple factors, including tumor heterogeneity, differences in engraftment efficiency, and individual host factors influencing disease progression and ctDNA shedding. Regarding translatability, while this variability presents challenges, it also mirrors the diversity observed in clinical practice. Our findings align with observations in clinical settings, where patients with seemingly similar disease classifications often exhibit varied responses and outcomes. This approach aligns with the growing emphasis on personalized medicine in oncology and the need for comprehensive, multi-faceted approaches to disease monitoring and treatment strategies in MLL-rearranged leukemias. We have expanded the Discussion to address this question (lines 426-433).
In the results there is analysis of the cell-free DNA but in the discussion, there is little mention of it or its comparison to the ctDNA findings.
Response: To address the Reviewer’s suggestion, we have included additional text in the Discussion section of the main text (lines 414-417).
Reviewer 2 Report
Comments and Suggestions for Authors
In this study, authors demonstrated the potential use of ctDNA as a biomarker for monitoring MRD and treatment response in a preclinical PDX model of pediatric MLL-rALL (n=3), that maintained the genetic characteristics of the patient derived cancer cells. The detection of ctDNA in these PDX models allowed longitudinal monitoring of the disease progression. Authors observed a correlation between % huCD45+ in the PB and ctDNA levels in plasma of these PDXs. Further, authors proposed ctDNA as a surrogate indicator of disease burden as compared to % huCD45+ in mouse PB. Finally, authors concluded “ctDNA levels accurately predicted the presence of MRD in the BM and spleen, providing a more comprehensive picture of treatment response than conventional flow cytometry methods.”
· Significant differences were observed in the quantity of cfDNA among 3 donors although clinical parameters of the patients were similar. Was there any correlation between cfDNA and ctNDA in respective PDXs? Did authors observe any correlation between cfDNA and %CD45+ cells? Fig2A showing cfDNA in MLL-1/2/7 wrt naïve mice showed 2-6 data points in scatter plot? Please explain what these data points show and why no.s are different for each PDX (n=5-6) / naïve mice (n=2)?
· The manuscript is well written, organized and clearly demonstrated results with figures and tables. Authors have identified the limitations of current study and discussed it in the manuscript. To support conclusion statement “providing a more comprehensive picture of treatment response” authors need to provide more supportive data. In its present form, ctDNA data provided in the manuscript does not support the predictive response to treatment with SNDX-50469.
Author Response
Significant differences were observed in the quantity of cfDNA among 3 donors although clinical parameters of the patients were similar. Was there any correlation between cfDNA and ctNDA in respective PDXs? Did authors observe any correlation between cfDNA and %CD45+ cells? Fig2A showing cfDNA in MLL-1/2/7 wrt naïve mice showed 2-6 data points in scatter plot? Please explain what these data points show and why no.s are different for each PDX (n=5-6) / naïve mice (n=2)?
Response: We did not observe significant correlations between cfDNA and ctDNA levels in the respective PDXs. This lack of correlation suggests that the total cfDNA quantity may not directly reflect the amount of ctDNA in these models. Interestingly, we did observe a significant correlation between cfDNA levels and %huCD45+ cells in the MLL-2 and MLL-7 PDXs. This data has been included in Supplementary Figure S2. The correlation suggests that in these models, the cfDNA levels may be indicative of the overall disease burden as represented by %huCD45+ cells.
As mentioned in the response to Reviewer #1 comment 2, Figure 2A shows 6 data points for each PDX model (MLL-1, MLL-2, and MLL-7), representing 6 individual mice engrafted with each PDX at 3 weeks post-inoculation. In this study, 6 engraftment was measured at three time points: 1, 2 and 3 weeks post-inoculation (all values shown in Figure 2B-D). Originally, 6 naïve mice were used as negative controls for this study and only 2 data points are shown as plasma was obtained from 2 naïve mice at this timepoint. The difference in sample numbers between PDX-engrafted and naïve mice is due to the allocation of 2 mice per timepoint.
The manuscript is well written, organized and clearly demonstrated results with figures and tables. Authors have identified the limitations of current study and discussed it in the manuscript. To support conclusion statement “providing a more comprehensive picture of treatment response” authors need to provide more supportive data. In its present form, ctDNA data provided in the manuscript does not support the predictive response to treatment with SNDX-50469.
Response: We agree that the relationship between ctDNA levels and MRD is more complex than our initial conclusion. In the case of the MLL-1 cohort and ctDNA detection in the absence of MRD in spleen and bone marrow, we suggest that the ctDNA levels may show MRD levels that are below the quantitative and sensitive ranges of conventional methods. This result is supported by evolving research suggesting that ctDNA detection may improve MRD assessment. We have updated our conclusion to more accurately address the relationship observed between ctDNA levels and MRD positivity (lines 386-389).
Reviewer 3 Report
Comments and Suggestions for Authors
Dear authors here are my suggestions:
Line 64: “needed to replace BM sampling” I think the verb replace its not correct, you should betetr use other word.
Line 65: “In recent years, the detection of cell-free DNA (cfDNA) and circulating tumor DNA (ctDNA) in plasma has garnered considerable attention as a minimally invasive modality”
Line 115: “sed to compare cfDNA and ctDNA”
In comparison the two previous comments, I believe you should explain more for cfDNA and ctDNA
Line 70: “while promising studies” you could put at this place some references
Line 105: “hours”. See the next comment
Line 138: “Hours” you should put abbreviation and be the same in all text
Line 230-232: “These findings highlight that in preclinical models, ctDNA allows for accurate longitudinal measurement of disease progression” So far what happened? Comparison in discussion section
Line 329: “as a biomarker” I believe its not easy to use, to characterize a parameter as biomarker. I believe you should have more evidence. You could refer as suggested biomarker
Line 352“biomarker” you should better write “possible biomarker” or “suggested”
Question1: The total gain from using the inhibitor menin in your study?
Question2: What is the State of the Art of your work?
Author Response
Line 64: “needed to replace BM sampling” I think the verb replace its not correct, you should betetr use other word.
Response: We agree with the Reviewer’s comment. Therefore, we have updated the text in the manuscript (lines 57, 62 and 64).
Line 65: “In recent years, the detection of cell-free DNA (cfDNA) and circulating tumor DNA (ctDNA) in plasma has garnered considerable attention as a minimally invasive modality”
Line 115: “sed to compare cfDNA and ctDNA”
In comparison the two previous comments, I believe you should explain more for cfDNA and ctDNA.
Response: Please see the response to Reviewer #1 comment 1.
Line 70: “while promising studies” you could put at this place some references.
Response: We appreciate the Reviewer’s suggestion. At the end of line 78, we have included references from a range of comprehensive reviews of ctDNA in adult and solid cancers.
Line 105: “hours”. See the next comment
Line 138: “Hours” you should put abbreviation and be the same in all text.
Response: We agree with the Reviewer’s suggestion and have updated the text in the manuscript accordingly (lines 110 and 147).
Line 230-232: “These findings highlight that in preclinical models, ctDNA allows for accurate longitudinal measurement of disease progression” So far what happened? Comparison in discussion section.
Response: We thank the Reviewer for this question to improve our manuscript. In the Discussion section, we have referenced similar studies for ctDNA assessment in other preclinical models and have revised the text to better reflect their findings (lines 393-394).
Line 329: “as a biomarker” I believe its not easy to use, to characterize a parameter as biomarker. I believe you should have more evidence. You could refer as suggested biomarker
Line 352“biomarker” you should better write “possible biomarker” or “suggested.”
Response: We appreciate the Reviewer’s suggestion and have improved our conclusions as shown in the updated text (lines 383, 407and 431-432).
Question1: The total gain from using the inhibitor menin in your study?
Response: We thank the Reviewer for this question. As our aim was to model the various clinical stages of MLL/KMT2A-r ALL including remission and relapse, we treated mice with SNDX-50469, which from our previous studies and other studies, showed outstanding efficacy and highly reproducible disease relapse or cure in several mice. This experimental model allowed us to assess ctDNA concentrations, provided an opportunity to quantify MRD and organ infiltration of known leukemia sites and understand the correlation between ctDNA with treatment response. The results gained from this study presented important proof‐of‐principle data and further support the application and feasibility of ctDNA analysis in preclinical studies.
Question2: What is the State of the Art of your work?
Response: We thank the Reviewer for this question. The current study addresses the challenges of conventional MRD methods (which include sensitivity determined by BM sample input and inability to predict relapse in extramedullary sites), by investigating the potential role of ctDNA analysis as a complementary tool for the assessment of disease burden in childhood leukemia. This is the first report of ctDNA dynamics in a clinically relevant experimental model of high-risk ALL. These models are essential for the translation of liquid biopsy-based molecular monitoring as an adjunct to establishing diagnosis and risk classification or assessing treatment response.
Round 2
Reviewer 3 Report
Comments and Suggestions for Authors
Dear authors
Thank you for your responses. Good job!